# Factors Influencing Smallholder Farmers' Decisions to Participate in Loan-Based Farming in Mutare District, Zimbabwe—A Double-Hurdle Model Approach

Tariro Mafirakurewa [1], Abbyssinia Mushunje [1] and Siphe Zantsi [2],*

[1] Department of Agricultural Economics and Extension, University of Fort Hare, Alice Campus, Alice 5700, South Africa; 201713660@ufh.ac.za (T.M.); amushunje@ufh.ac.za (A.M.)

[2] Economic Analysis Unit, Agricultural Research Council, 1134 Park Street, Hatfield, Pretoria 0028, South Africa

* Correspondence: siphezantsi@yahoo.com

**Abstract:** Agriculture is an important sector in Zimbabwe's economy. More than 70% of the population are smallholders relying on agriculture. To support agriculture, Zimbabwe's government introduced a Targeted Command Agriculture Programme (TCAP), in terms of which the state provides production inputs like seeds, fertilisers, protection chemicals, and extension services. In turn, the farmer is expected to produce 5 tonnes of maize per hectare for the Grain Marketing Board. The cost of inputs that the state provided is then deducted from the 5-tonne maize yield, and the farmer is paid the balance. Numerous authors have studied the design of TCAP and its impact on farmers. However, only a few have focused on the determinants of participation, especially with an empirical basis and in the Mutare District. To bridge this knowledge gap, this study implements a double-hurdle model to determine factors influencing farmers' participation in Zimbabwe's TCAP using a sample of 350 farmers. The study found that gender, family size, farmer type, command agriculture education, and distance from the market influenced smallholder farmers' participation in TCAP. Therefore, policymakers should consider these factors to improve the design of the programme and enhance the participation of smallholder farmers in it.

**Keywords:** command agriculture; contract farming; agricultural financing; rural development

## 1. Introduction

Agriculture plays a central role in the economies of low-income nations such as Zimbabwe. Nevertheless, smallholder farmers face a severe financial shortage that limits the purchase of productive agricultural inputs. Similar to other developing nations, access to agricultural credit is crucial in Zimbabwe to enhance the well-being of these farmers. For example, smallholder farmers in Zambia [1,2], Pakistan [3], Kenya [4], and India [5] face many challenges in accessing financial services. Zimbabwe has been making an effort to enhance farmers' productive capacity by providing finance and investing in their physical and human capital since independence in 1980.

Since independence, Zimbabwe's agricultural sector has undergone several structural changes, especially due to the land reform programmes implemented in the country. Deininger et al. [6] highlight the following incidents when describing Zimbabwe's land reform history: For many years after independence, the government of Zimbabwe was committed to redressing the historical injustices by acquiring farmland on a willing buyer, willing seller basis. The approach worked well, exceeding the targets in some instances; for example, by the end of 1989, 3.5 million hectares of land had been acquired, and 71,000 households had been resettled—more than the original target of 18,000 but far from the subsequently revised target of 162,000 households. A year later, the constitutional obligation to proceed on a willing buyer, willing seller basis expired and was succeeded by

the Land Acquisition Act (a new form of acquisition of farms). Land reform then proceeded at a very slow pace until the late 1990s.

The liberation fighters (war veterans) were displeased with the slow pace of the land reform programme and teamed up with angry citizens in land grabbing and farm invasions with no compensation [7,8]. This unplanned and erratic policy paradigm shift marked the beginning of the Fast-Track Land Reform Programme (FTLRP started in 2000), where the Zimbabwean government repossessed vast tracts of land. Mkodzongi and Lawrence [9] argue that, since 1998, the programme has effectively co-opted farm occupations, redistributing land from white-owned farms, estates, and state holdings to over 150,000 farmers using two models: A1 and A2. The A1 model allocates small plots for growing crops and grazing land to landless and poor households. In contrast, the A2 model allocates farms to new black commercial farmers with farming skills and resources to farm profitably and to reinvest and increase agricultural productivity [10,11].

Agricultural production post-FTLRP continues to be affected by a myriad of challenges, including poor access to and availability of inputs, vulnerability to weather-related shocks like droughts and floods, pests, and poor quality (especially in communal areas), and a lack of credit lines [12]. Because of the lack of safe collateral and the dangers associated with insecure tenure and selective law enforcement, commercial banks have refused to provide loans to A1 and large-scale farmers since the beginning of Zimbabwe's land reform programme. Banks also refuse to accept the government's 99-year leases, which have to be more secure and non-transferable [11], making the arrangements unsuitable for collateral purposes.

Other challenges include underdeveloped agri-food value chains, which significantly increase the bank's risk and exposure, and a general need among financial institutions to understand the agricultural sector and its opportunities. This has led to low financing of smallholder farmers, low cereal crop production, and reliance on maize imports. Commercial banks prefer to provide loans to large, established businesses rather than small loans to numerous micro-entrepreneurs [13]. As a result, a yawning smallholder farmer financing gap means that Zimbabwe's agricultural potential remains untapped.

Faced with deteriorating agricultural production, unprecedented inflation, and the collapse of the economy, Zimbabwe adopted contract farming, namely the Targeted Command Agriculture Programme (TCAP), as a financing model to boost agricultural production. The TCAP aimed to cut the country's food import costs, beginning with the 2016/2017 crop season [13]. As defined by Li and Stuber [14], contract farming is an agricultural production system based on an agreement between buyers and farmers that specifies conditions for the production and distribution of farm products. The TCAP was designed to fill the vacuum in the financing of the production of cereal grains [15]. Under the TCAP, the state provides inputs like seeds, fertilisers, protection chemicals, and extension services. At the same time, the farmer is expected to deliver 5 tonnes of maize per hectare to the Grain Marketing Board (GMB) (government parastatal) [15]. The cost of the inputs the state provides is deducted from the maize delivered, and the farmer is paid the balance on the loan [13].

To date, several studies have been conducted to understand TCAP and its impact on farmers. Among these studies, a majority have attempted to understand TCAP in general. These include studies by Mazwi et al. [13], Odunze and Uwizeyimana [16], Nkala [17], and the World Bank [18]. Makuwerere [12] studied TCAP's impact on food security, and Shonhe and Scoones [19] researched contract farming in Zimbabwe by focusing on accumulation, social differentiation, and rural politics. However, few empirical studies have been conducted on the determinants of TCAP participation among maize farmers in the Mutare District. Therefore, this study is an attempt to determine the factors influencing farmers' participation in TCAP. This objective was achieved empirically by implementing a double-hurdle econometric model to a sample of 350 farmers in the Mutare District.

A survey of the available literature on the factors influencing smallholder farmers' participation in contract farming revealed that several authors, including Muroiwa et al. [20], Khan et al. [21], and Taslim et al. [22], used binary logistic regression, while Etwire et al. [23]

employed the binary probit model and Gaisina [24] utilised the bivariate probit model. However, all the analytical methods above only provide a binary outcome on the factors that influence the farmers' participation in contract farming and do not look into the factors that influence the extent of participation. To close this gap, this study employed the double-hurdle regression model, mainly because it allows for a more comprehensive analysis of both the decision-making process leading up to contract participation and the intensity of participation in contract farming. The double-hurdle regression model also accounts for unobserved heterogeneity by treating participation and intensity as separate decisions. This provides a more nuanced understanding of the factors that influence farmers' decisions to participate in contract farming.

This study unfolds as follows: Section 2 presents a theory that grounds the present study. Section 3 describes the study site, the dataset, and the methods employed to achieve the objective of the study. Section 4 presents and discusses the results, and Section 5 concludes and offers some recommendations for policymakers.

## 2. The Theory of Demand

The study is grounded in demand theory, which was developed by Adam Smith in 1776 [25]. The theory specifies the relationship between the demand for and price of services and goods. The demand for loans is equivalent to the demand for other services. Ben-Akiva and Lerman [26] assert that the prime determinant of the demand for credit is the interest rate, which, in this case, is regarded as the price. Like the demand curves of other goods, the demand curve of interest rates slopes downwards [27], as shown by $D_1$ and $D_2$ in Figure 1 below. An increase in interest rates results in a decline in the demand for credit services. Zimbabwe is characterised by a perfectly competitive financial market, implying that market forces determine the best rates. In reference to McConnel [28], the interest rate is offered when credit demand equates to credit supply. The four basic laws of demand and supply state, firstly, that increasing demand and holding supply constant creates a shortage, thereby increasing equilibrium prices. Secondly, a decrease in demand and a constant supply result in a surplus, resulting in a decline in equilibrium prices. Thirdly, a constant demand and a rise in supply create a surplus and decrease equilibrium prices. Lastly, a constant demand and an increase in supply result in a shortage, raising the equilibrium price.

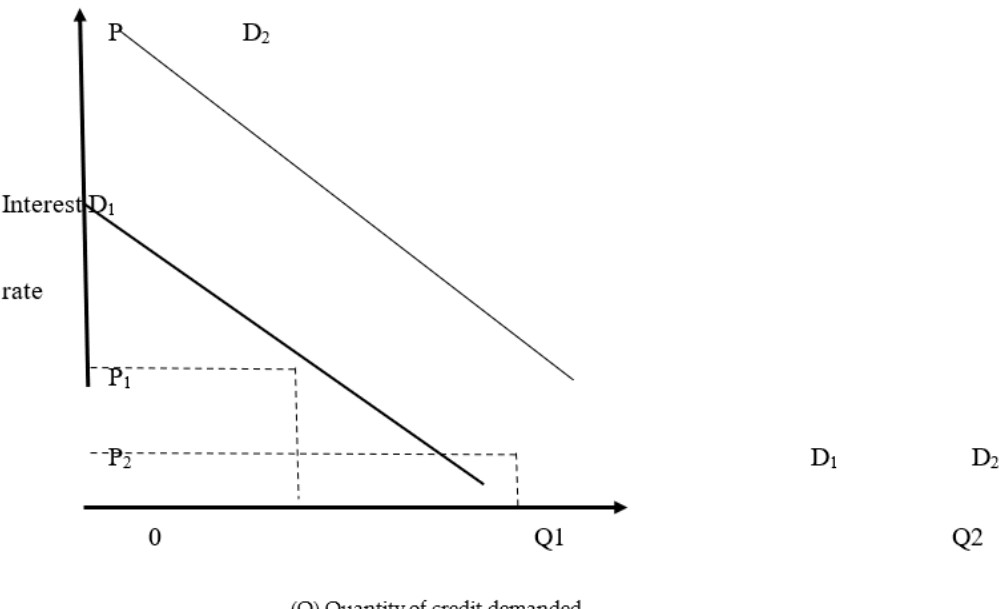

(Q) Quantity of credit demanded.

**Figure 1.** The demand curve for agricultural credit. (Source: own computation).

The credit market represents total demand by interest rates, given the available information, customer service, and credit alternatives. The vital factors affecting credit demand in the Zimbabwean credit market include varieties, customer information, and customer service [26]. Like all normal goods, the marginal cost of credit is the supply curve for credit by the government of Zimbabwe, and the marginal utility curve is determined by the demand curve [27]. In agriculture, farmers are keen to take credit at a particular interest rate if the marginal utility of the credit is equivalent to the opportunity cost. Hildenbrand [28] asserts that the opportunity cost is defined by the price, which is the marginal utility of alternate credit sources like microfinance and banking institutions.

In this study, the ability and willingness of a farmer to take a loan from the government at a given time represent the demand schedule. Loan uptake can be expressed by changing demand theory to develop the following model: $C = f(=)$, where the vector of credit uptake factors is depicted by $(=)$. Njiru [29] adds that the cost of credit is the main obstacle to accessing credit. High interest rates or borrowing costs discourage farmers from taking loans from formal institutions. Figure 1 illustrates the relationship between interest rates and the quantity of credit demand at different interest rates.

In Figure 1, the y-axis represents the interest rate, which is the price of credit (P), and the x-axis represents the quantity of agricultural credit demanded by farmers. The demand curves ($D_1$ and $D_2$) are downward sloping, indicating the inverse relationship between the interest rate and the quantity of credit demanded by farmers. At the interest rate (Price) $P_1$, the quantity of credit demanded is $Q_1$, while $Q_2$ represents the quantity of credit demanded at interest rate $P_2$. The difference between $Q_1$ and $Q_2$ ($Q_1 < Q_2$) shows that farmers are willing to borrow more credit at a lower interest rate.

## 3. Methodology

### 3.1. Study Area

Manicaland is a province of Zimbabwe that borders Mozambique (Figure 2). It has seven districts: Chimanimani, Buhera, Mutare, Mutasa, Makoni, Nyanga, and Chipinge. The Fast-Track Land Reform Programme allocated 273,176.87 hectares to 13,000 households in both the A1 and A2 resettlement models, with the Mutare District having 1553 A1 and 158 A2 farmers [30]. The Mutare District, which is located in natural farming regions 3 and 4, was selected for this study because of the high concentration of resettled farmers participating in command agriculture maize production, which provided enough data to make the study feasible and robust. The district receives an average annual rainfall of 600 mm to 800 mm [31].

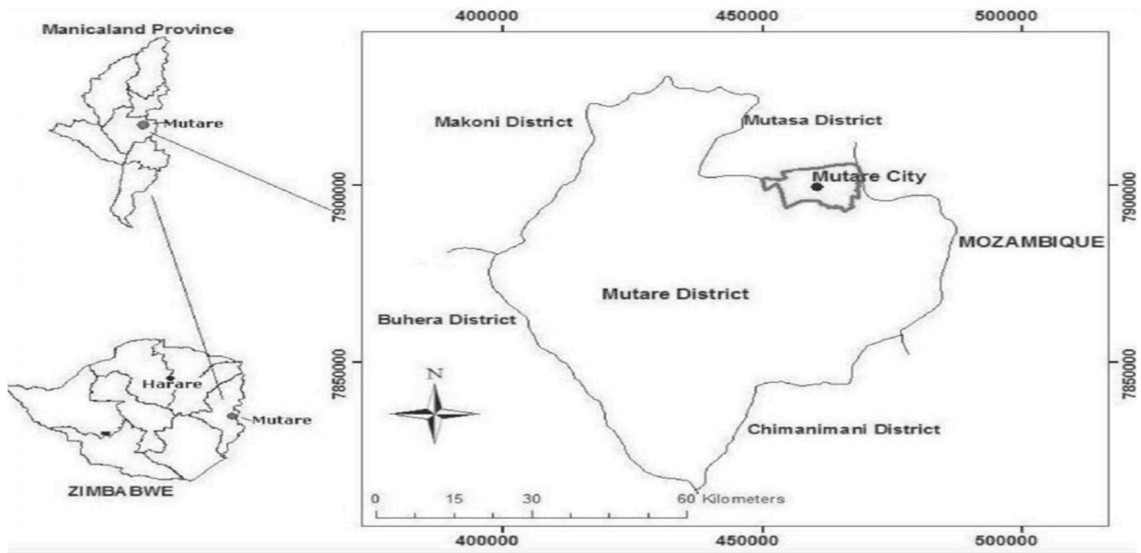

**Figure 2.** Map of the Mutare District. (Source: Google Maps).

Maize is cultivated extensively in the district, as it serves as the country's staple crop and is an integral part of smallholder farmers' produce. The cultivation of maize in the district typically commences after the late-October rains. Farmers also grow other crops, such as sorghum, millet, wheat, beans, ground nuts, tobacco, round nuts, and horticultural produce, to diversify their agricultural production. In addition to agricultural production, households in the study region engage in animal production, with cattle, rabbits, goats, sheep, piglets, and broilers reared the most commonly. Livestock rearing generates income while contributing to food consumption in the study area.

*3.2. Sampling Procedure and Sample Size Selection*

The target population for this study was the A1 and A2 resettlement farmers in the study area. The study employed a multistage sampling technique. Firstly, there was the purposive selection of Manicaland province, given that it is endowed with the five natural farming regions in Zimbabwe. Secondly, there was a purposive selection of the Mutare District among the four districts with favourable climatic conditions, given time, and financial constraints. While the last stage involved stratified sampling, the sample was divided into two strata. Stratum 1 had A1 farmers, and stratum 2 consisted of A2 farmers; a random sample was selected from each stratum. The reason behind selecting these two categories was to avoid bias and ensure that each group of resettled farmers was fully represented. The following formula by Kothari [32] was used to determine the sample size.

$$n = \frac{Z^2 pqN}{(N-1)e^2 + Z^2 pq},$$

where n = sample size, $N$ = population size and $e$ = level of precision (5%), $Z^2$ = confidence level (95% = 1.96), $p$ = the population proportion (assumed to be 0.5, for it provides the maximum sample size), and $q = (1 - p)$.

Calculating the sample size of A1 farmers who participated in command agriculture,

$$n = \frac{1.96^2 (0.5)(1-0.5)(685)}{(685-1)\, 0.05^2 + 1.96^2 (0.5)(1-0.5)} = 250,$$

therefore, 250 A1 farmers were supposed to be sampled for A1 farmers.

$$n = \frac{1.96^2 (0.5)(1-0.5)(76)}{(76-1)0.05^2 + 1.96^2 (0.5)(1-0.5)} = 70,$$

Calculating the sample size of A2 farmers participating in command agriculture,

The study involved 250 A1 and 70 A2 farmers participating in command farming. In addition to the above, the counterfactual group was involved in analysing the factors influencing loan uptake and repayment among A1 and A2 resettled smallholder farmers. Although many authors argue that the treatment and control groups should be evenly split [33,34], Oldfield [35] finds that this assumption is false and that using unequal groups may actually gain more power in the study. For example, unequal samples can be used due to time and budget constraints [35,36]. In this study, 125 A1 and 40 A2 farmers were selected as control groups. Generally, the study involved 320 farmers who participated in command agriculture and 160 farmers from the counterfactual group (non-participants in command agriculture maize production), making a total sample of 480 farmers.

Data were collected through personal interviews, guided by a semi-structured questionnaire, between June and August 2022. To gather relevant information, the questionnaires contained both open-ended and close-ended questions because open-ended questions. Firstly, open-ended questions were employed because they allow respondents to express their feelings, thoughts, and experiences in their own words [37]. Secondly, open-ended questions do not limit respondents to predefined options, giving the researchers the potential to uncover unexpected insights and perspectives that might not have been considered otherwise [37].

This can lead to the discovery of novel findings and original contributions to the field. On the other hand, the close-ended questions increased the robustness of the questionnaire by ensuring that all respondents answered the same set of options (standardised responses). This consistency allows for easier comparison between different respondents or groups [38]. Furthermore, closed-ended questions help reduce bias by ensuring that all respondents receive the same set of response options and provide a standardised survey experience [39,40]. After the data-cleaning process, a total of 350 questionnaires were suitable for analysis, constituting 73% of the intended sample size.

### 3.3. Data Analysis

The double-hurdle regression model [37], which considers sample-selection bias (due to the non-random decision of individuals (farmers) to participate in contract farming), was implemented to determine factors that influence smallholder farmers' participation in command agriculture. The model assumes that the borrowing process of individual farmers involves two sequential stages [38], and a set of factors determines each stage. Corresponding to the behavioural content of this model, individuals must pass two separate hurdles before obtaining credit.

In the first hurdle, the decision on whether or not to participate in contract farming (participation decision) is made. This is mainly because the decision to participate lies with the farmer based on utility to maximisation, which is further limited by the contractor/contract conditions. Just like in advertising, an influence is made, but the consumer makes a decision at the end. The contract conditions can be an independent variable; however, that was not per view of the study. The second hurdle involves the extent of participation (loan sizes) in the loan obtained by farmers, which may be affected by several factors related to individual farmers, financial institutions' characteristics, etc.

The two decisions can be regressed as dependent on or independent of each other [40]. This allows for the use of the same or different determinants in each of the two decision equation hurdles. In this case, the hurdles were run simultaneously. The reason for this is that modelling correlated decisions in the first and second stages, such as factors influencing participation and the extent of participation, provides a more accurate representation of the decision-making process. The joint modelling of hurdles enhances efficiency and precision, particularly with shared explanatory variables, resulting in more reliable parameter estimates and improved predictions. It estimates a probit equation (the probability of receiving a loan in our case) and a pooled OLS (for the loan amount in our case), including the Mills ratio. The inverse Mills ratio, a selectivity term in the selection model, is incorporated into the outcome equation to correct sample-selection bias [41]. Following Greene [42], the hurdle equation is expressed as follows:

$$W^* = \alpha' Z + u, \; Z_i^* = \alpha_i L_i + u_i \tag{1}$$

Equation (1) measures the probability that a household *i* has access to formal credit; $Zi^*$ is a dummy that a household takes a loan; and $L_i$ is a vector of exogenous household variables that affect $Zi^*$. The variable $Zi^*$ is not observed, but we observe if the household has access to credit or not, whereby $Zi = 1$ if $Zi^* > 0$ and $Zi = 0$ if $Zi^* \leq 0$.

Next, household characteristics are also assumed to influence the size of the loan the households take up. In this case, the above statement on household characteristics reads to the contrary of the statement in the previous paragraph, which states that $L_i$ is a vector of exogenous household variables that affect $Z_i^*$ Under the condition that $Z_i = 1$, Yi represents the log of the loan size expected to be received by each household, with the following assumption:

$$Y_i = b_i X_j + v_i, \tag{2}$$

where $X_i$ is the vector of variables determining the loan size. In Equations (1) and (2), ui and vi follow bivariate normal distributions with zero means, standard deviation $\delta u$, and $\delta v$, correlated with correlation coefficient ρ. It is assumed that $Z_i$ and $L_i$ are observed for a

random sample of farmers, but $Y_i$ is observed only when $Z_i = 1$ when farmer *i* has access to command agriculture loans (credit). Then

$$E(\frac{Y_i}{C_i} = 1) = E(Y_i/C_i^* => 0) = E(Y_i/ц_i > -y'z_i = \beta'x_i + E(\varepsilon_i/ц_i > -y'z_i) = \beta^i x_i > \rho\delta\varepsilon\lambda_i(\alpha ц_i), \tag{3}$$

where $\lambda_i(\alpha u) = \frac{\theta(\alpha u)}{1-\phi(\alpha u)} = \frac{\theta(-\alpha u)}{\phi(-\alpha u)} = \frac{\theta(Y'zi/\sigma u)}{\phi(Y'zi/\sigma u)}$, ø is the normal density function and Φ is the normal distribution function, while the function $\lambda_i(\alpha u)$ is the inverse Mill's ratio. The least squares regression of $Y_i$ on $x_i$, omitting the term in $\lambda_i(\alpha u)$, would produce an inconsistent estimator of $\beta$ because of the correlation between $x_i$ and $\lambda_i(\alpha u)$. To avoid inconsistency, if the expected value of the error was known, it could be included in the regression as an explanatory variable to eliminate the part of the error correlated with the explanatory variables [43]. In Heckman's model, the first stage estimates the expected error value, and the second stage uses it as an additional variable in the regression equation. The probit model estimates parameters $\gamma$ of the $Z_i$ equation by maximum likelihood. The estimates of $\gamma$, $\lambda$ for each observation are used as an exogenous variable in the $Y$ equation, allowing the least-square regression of $Y_i$ on $x$ and $\lambda$ to estimate parameters consistently $\beta$.

This study hypothesises that some socio-economic factors significantly affect individual farmers' decisions and the extent of their participation in contract farming. The variables specified in the models are presented in Equation (4) below.

$$Z_i = \beta_0 + \beta_1 \text{AGE} + \beta_2 \text{FAMLYSIZ} + \beta_3 \text{DMKT} \ldots\ldots\ldots\ldots\beta_i X_j \tag{4}$$

Table 1 below, informed by the literature in the next paragraph, tabulates the description of the variables used in the analysis as well as their expected influence in the double-hurdle models.

**Table 1.** Definition and expected signs for explanatory variables used in the analysis.

| Variable Name | Description | Expected Sign |
| --- | --- | --- |
| Gender of the farmer | 1 = male, 0 = female | − |
| Farmer type | 0 = A1, 1 = A2 | − |
| Educational level | 0 = secondary and below, 1 = certificate/ diploma or above | + |
| Family size | 1 (5 and below), 2 (6–10), 3 (10 and above) | + |
| Training | Training on command agriculture loans | + |
| Distance of farm from the loan-issuing office | Kilometres | − |

### 3.4. Choice of Variables Used in the Empirical Analysis and Justification for Inclusion

Gender was included in the model as a dummy variable in Zimbabwe due to women's lower decision-making power in credit participation in rural areas. Chandio [44] reveals that male farmers possess more resources like land, which act as collateral security for accessing formal credit compared to their female counterparts. The study suggested that male respondents have a higher demand for formal credit, suggesting that gender positively influences farmers' participation in command agriculture.

Education was also measured as a dummy (0 = secondary and below, 1 = tertiary) and was included in the model because of its significant influences on access to and control of productive resources, including loans, and is a key factor in the decision to acquire and participate in command agriculture loans [45]. The acquisition of higher educational levels was expected to increase farmers' financial literacy, which is associated with decision making and efficient credit application, and increase the possibility of farmers participating in the loan [46].

Family size was measured in three categories (1 = less than 5, 2 = 5–10, and 3 = 11+) because farmers with large families are more likely to participate in command farming than those with small families due to having available farm labour from family members, which reduce the production costs, increase profit, and increase loan repayment possibilities *ceteris paribus* [47]. The consumption requirements of a household increase as household size increases, and this results in increased stress on available limited resources. According to [48], households with more family members demand more credit. Therefore, this study hypothesises that family size positively influences farmers' participation in command agriculture.

Distance from GMB (market) was measured as the actual distance from the collection points in kilometres. This variable was important because farmers are likely to produce more if they are near the market compared to distant ones because of easy access and affordable output transport costs at shorter distances.

Farmer type was measured as a dummy variable with 1 = A1 farmer and 2 = A2 farmer. The study anticipated that A2 farmers would participate more and obtain larger loans for large maize production since they have more land (36 hectares or more) than A1 farmers, who possess less than 10 hectares, with the majority owning only 5 hectares. According to [49], landholding sizes greatly influence loan participation.

The loan size was measured as a continuous variable and was used to measure the extent of participation. This variable was included in the model because loan sizes determine the profit of borrowers, with larger loans increasing the profit of borrowers [50] and vice versa. Training on command agriculture loan enhances resettled farmers' financial literacy, boost their confidence, and nurtures their entrepreneurial skills, making farmers more profitable and appealing to lenders.

## 4. Results and Discussion

### 4.1. Description of Sample Characteristics

Table 1 presents the descriptions of the sample, with results expressed in percentages due to the categorical nature of the variables, making it impossible to calculate the mean and standard deviation for each variable. The results reveal that 74% of farmers in the study area are male. This is not surprising because the Fast-Track Land Reform Programme favours men with more land allocation than women in all provinces [41]. While female farmers actively participate in various farming tasks [42–44], this study revealed that they do not own the land they work on. Therefore, their ability to access credit depends on their relationship with their husbands (who own the land).

Most A1 and A2 farmers are in the middle-age group (as shown in Table 2). This indicates that the majority of farmers are in the productive age range, capable of strenuous tasks to repay loans. Chandio and Jiang [45] state that formal credit favours young, educated farmers for efficient resource allocation through technology adoption. In terms of educational level, more A2 farmers had tertiary education compared to A1 farmers, who mostly had a secondary education as their highest education level.

**Table 2.** Summary of the demographic information of the sample (N = 350).

| Variable | Description | A1 | A2 |
|---|---|---|---|
| Gender | Male | 71% | 90% |
| | Female | 29% | 10% |
| Age | <35 | 10% | 2% |
| | 36–50 | 49% | 27% |
| | 50 | 41% | 71.% |
| Family size | 1–5 | 41% | 69% |
| | 6–10 | 47% | 26% |
| | 11+ | 12% | 45% |

**Table 2.** *Cont.*

| Variable | Description | A1 | A2 |
|---|---|---|---|
| Educational level | Secondary | 96% | 45% |
|  | Tertiary | 4% | 55% |

*4.2. Farm Characteristics*

Table 3 summarises the farm characteristics of A1 and A2 farmers, showing percentages and averages of farmers owning the various farm equipment and the standard deviation. The standard deviation measures the dispersion of values around the mean, indicating how much a dataset's values vary from the average value. A low standard deviation indicates that values are close to the mean, while a high standard deviation indicates that values are spread out. This quantifies how much the values in a dataset vary from the average value.

**Table 3.** Summary of AI and A2 farm characteristics of the sample (n = 350).

| Variable | Description | A1 n = 301 | Mean | Std Dev. | A2 n = 49 | Mean | Std Dev. |
|---|---|---|---|---|---|---|---|
| Total land | Hectares | 1647 | 5.57 | 2.03 | 6826 | 139.3 | 79.60 |
| Total arable land | Hectares | 1604.8 | 5.33 | 1.42 | 3773 | 77 | 52.32 |
| Type of farming | Dryland | 99% |  |  | 94% |  |  |
|  | Irrigation | 0.1% |  |  | 0.06% |  |  |
| Farm equipment | Ox-drawn plough | 98% | 0.98 | 0.13 | 78% | 0.80 | 0.39 |
|  | Maize storage house | 99% | 0.98 | 0.11 | 96% | 0.96 | 0.18 |
|  | Tractor and tractor implements | 0.017% | 0.02 | 0.12 | 39% | 0.33 | 0.47 |
| Livestock | Cattle | 82% | 0.81 | 0.38 | 90% | 0.89 | 0.30 |
|  | Goats | 80% | 0.79 | 0.39 | 90% | 0.89 | 0.30 |
|  | other | 98% | 0.98 | 0.34 | 96% | 0.95 | 0.27 |
| Crops grown | Maize | 100% | 1.64 | 0.92 | 100% | 7.45 | 12.05 |
|  | Tobacco | 49% | 0.64 | 0.83 | 45% | 3.39 | 10.1958 |
|  | Other | 77% | 0.50 | 0.43 | 51% | 2.27 | 3.65 |

(Source: survey data).

Our results in Table 3 show that A2 farmers owned larger tracts of land compared to A1 farmers despite being a large part of the sample. This is mainly because the A1 model allocated small plots for growing crops and grazing land to poor and landless farmers [9]. At the same time, the A2 model allocated farms to new black commercial farmers who were supposed to have the skills and resources to reinvest, farm profitably, and raise agricultural productivity. Although there is much variation in land sizes between the A1 and A2 farmers, the average size of new A2 farms is 318 hectares, while that of A1 family farms is 37 hectares, including crop and grazing land [46].

This study reveals that, among the total land (8,473 hectares) owned by A1 and A2 farmers, 1,542.83 hectares were allocated to crop production. The A2 farmers had a total arable land area of 3773 hectares, and 3289.3 hectares were used for crop production, while the remaining 483.97 hectares were reserved for animal grazing land. This shows that the A2 farmers utilise a large proportion of their arable land. A1 farmers owned 1647 hectares, with 1604 hectares of arable land. A total of 840.3 hectares were used for crop production, while 806.7 hectares were left for grazing land.

In addition, most A1 (99%) and A2 farmers (94%) practiced dryland farming. Regarding farm equipment, most farmers in both farmer categories owned an ox-drawn plough, implying that the farmers in the study area rely heavily on draft power. Furthermore, 99% of A1 and 96% of A2 farmers had a maize storage house. Furthermore, 39% of the A2 and less than 1% of A1 farmers owned tractors and tractor implements, with a mean

of 0.017 and 0.34 for the A1 and A2 farmers, respectively. The standard deviation is 0.12 and 0.47, respectively. Also, the A2 farmers owned ridgers (31%), trailers (37%), and disc ploughs (33%), which were owned by fewer than 1% of A1 farmers. This reveals that A1 farmers are still developing, asset deprived, and have minimum investments in their farming operations. A1 farmers are historically disadvantaged, with limited resources, and face financial constraints, limiting access to expensive mechanised equipment like tractors because they cannot afford to buy and maintain such equipment. Furthermore, the limited farm investment by A1 farmers results from their lack of creditworthiness to secure loans from financial institutions, and the advent of command agricultural loans is their big stepping stone.

In terms of livestock, the results show that a large proportion of both A1 and A2 farmers owned cattle, goats, and other livestock, with A2 farmers, given their land sizes, having more livestock numbers than A1 farmers. Apart from cattle, farmers owned livestock such as donkeys, sheep, chickens, and pigs. Both groups of farmers can sell livestock for revenue, which acts as a buffer against fluctuations in crop yields and market prices.

Both the A1 and A2 farmers in the sample grew maize. This is because maize is a staple crop crucial for food security and income generation in Zimbabwe [47]. Also, more than 90% of A1 farmers were growing other crops in conjunction with maize, while only 51% of A2 farmers practiced crop diversification. This is because A2 farmers specialised in specific high-value crops (tobacco, millet, soybean, sugar beans, etc.) for profit maximisation, and A1 farmers grew various crops to meet their households' diverse food needs.

### 4.3. Empirical Results

4.3.1. Factors Influencing Farmers' Decisions to Participate in Command Agriculture

To analyse the factors influencing farmers' participation in command agricultural loans, the double-hurdle regression was employed. The primary objective of the study was to determine factors influencing farmers' participation in command agriculture loans, and it was not a comparative study on the factors affecting participation between the A1 and A2 resettled farmers. It is important to note that the internal validity was informed by the literature as affecting the acquisition of loans and loan sizes. The inclusion of farm size as an independent variable was omitted due to concerns about the potential introduction of multicollinearity.

The results of the double-hurdle model are presented in Table 4, with column 2 showing the results from the first hurdle, which used a probit estimator (MLE) to estimate the factors influencing farmers' participation in command agriculture, and column 3 shows the results from the second hurdle, which were obtained by fitting the MLE to the truncated normal regression to estimate the factors influencing the extent of farmers' participation in command agriculture. The second hurdle in Table 4 was not measured in hectares but rather in the loan sizes acquired by farmers. The study followed the hurdles performed by [51,52], who did not separate the models for each farmer type but instead included a variable that captures farmer type as we did in this study. Further, due to uneven and smaller sample sizes, if we had to split the farmers (A1 and A2), that would. The model was significant at the 1% level.

Of the six explanatory variables, the maximum likelihood estimates of the probit regression found only two variables that significantly influenced farmers' participation in command agriculture in the study area. The results indicate that family size and distance to the Grain Marketing Board (GMB) were significant factors in participating in command agriculture at the 10% level.

**Family size:** As was expected, family size positively influenced farmers' participation in command agriculture loans, with a coefficient of 0.21. This means a unit increase in family size increases the likelihood of farmers' participation in command agriculture by 21%. The result shows that, as family size increases, there is a higher probability of farmers participating in command agriculture.

**Table 4.** Factors influencing resettled farmers' participation in command agriculture.

| Variable | Hurdle 1—Participate in Command Agriculture (Decision Model) | | | | Hurdle 2—Loan Size (Extend Model) | | | |
| --- | --- | --- | --- | --- | --- | --- | --- | --- |
|  | **B** | **Std. Error** | **z** | **P > \|z\|** | **B** | **Std. Error** | **z** | **P > \|z\|** |
| Gender | −0.30 | 0.25 | −1.17 | 0.24 | −0.43 * | 0.12 | −1.69 | 0.09 |
| Family size | 0.21 * | 0.12 | 1.75 | 0.08 | 0.33 *** | 0.01 | 2.54 | 0.01 |
| Farm type | −0.56 | 0.43 | −1.30 | 0.19 | −0.72 * | 0.25 | −1.91 | 0.06 |
| Educational level | 0.12 | 0.41 | 0.30 | 0.76 | 0.18 | 0.13 | 0.45 | 0.65 |
| Received training on command agriculture | −0.15 | 0.13 | −1.15 | 0.25 | −0.24 * | 0.38 | −1.92 | 0.06 |
| Distance to GMB | 0.01 * | 0.01 | 1.71 | 0.09 | 0.01 *** | 0.40 | 2.51 | 0.01 |
| | | | | **Model summary** | | | | |
| Mills lambda | 1.29 | 0.33 | 3.88 | 0.00 | | | | |
| Rho | | 1.39 | | | | | | |
| Sigma | | 0.93 | | | | | | |
| Chi-square | | 16.86 | | | | | | |
| Sig. | | 0.01 | | | | | | |

Significant at * = 10%, and *** = 1%, source: own calculations.

These findings align with the findings of [20,48], who discovered that the size of the labour force influences farmers' decisions to participate in contract farming. This is mainly because smallholder farmers rely on family members as a source of labour, which reduces their production costs and, in turn, increases their probability of participating in contract farming and the repayment of contract farming loans. It is evident from Table 1 that a large proportion of A1 farmers had family sizes ranging between 6 and 10 members. In contrast, most A2 farmers had family sizes ranging from one to five members, mainly because they were resource-endowed and could hire labour for maize production.

**Distance from the market:** The results showed a positive significant relationship between distance from the markets and farmers' participation in command agriculture, with a coefficient of 0.01. This implies that a unit increase in distance from the market is likely to increase the probability of farmers participating in command agriculture by one percent. It is interesting to note that farmers further from the market are more likely to participate in command agriculture than households closer to the market.

These findings concur with the findings of [49], who discovered a significant positive relationship between participating in contract farming and distance from the market. The TCAP links farmers to the GMB; therefore, the increase in farmers' participation, as distance from the market, can be attributed to the reduced transaction costs mainly incurred during a search for markets. Command agriculture organisers the GMB to establish seasonal depots near farmers and buy maize at predetermined prices. Unlike farmers located far from the market, the farmers located near the market are highly exposed to spot buyers and or private buyers who buy their maize at their farm gates.

4.3.2. Factors Influencing Farmers' Extent of Participation in Contract Farming

The second hurdle model in Table 3 shows the maximum likelihood estimates of the truncated regression model for factors influencing the extent of farmers' participation in command agriculture loans. The factors significantly influencing the extent of participation in command agriculture include gender, farm type, command agriculture training (10% level), family size, and distance to GMB (1% level).

**Family size:** As was expected, family size was proven to have a positive significant relationship with loan size. The coefficient for family size is 0.33, which implies that a unit increase in family size increases the likelihood of farmers increasing their loan sizes by 33%. The result reveals that large-sized families were likely to acquire larger loan sizes.

The findings are similar to those of [50], who discovered that the dependency ratio of farmers also influences their uptake of formal loans. Since smallholder farmers grow crops

for family consumption and to ensure family food security and generate income (profits), farmers with large families tend to obtain large loans since most of the farmers in the study depended on agriculture as their primary source of income.

Among the participants, A1 farmers opted for substantial loan sizes to cultivate maize, primarily for family consumption. The surplus maize they produce is then sold, generating additional income to support their families. On the other hand, A2 farmers participate in command agriculture maize farming, with the primary objective of maximising profits.

**Distance from the market (GMB):** Unexpectedly, a positive relationship was found between distance to the GMB and loan size. The coefficient was 0.01, which implies that the size of the loan was significantly higher as the distance to the GMB increased. These results reveal that a unit increase in distance from the market (GMB) increases the likelihood of farmers increasing the loan size by one percent. These findings reveal that farmers located far from the market are using contract farming as a stepping stone to boost their maize production, food security, and incomes since most of them rely on agriculture as a source of income. Farmers close to the city, where the GMB is located, have options to sell directly to city consumers, unlike those far away who rely mostly on the GMB market. Above all, the increase in loan sizes as the distance from the market increases can be attributed to the increase in land sizes as we move away from the city; therefore, farmers demand more credit due to abundant land for farming.

**Gender:** The results showed a negative significant relationship between gender and loan size, with a coefficient of −0.43. This implies that male farmers were less likely to have large loan sizes than female farmers. These findings are contrary to [53], who discovered that there is no difference between average loans for male and female farmers, and [52,53], who found that men receive higher average loans compared to women. The study revealed that adding one female farmer among the command agriculture participants increased the likelihood of female farmers obtaining larger loans than males by 12%, ceteris paribus. This indicates that command agriculture indirectly empowers women, who now have a greater risk tolerance, thereby reducing their vulnerability to poverty and utilising credit to empower their family concerns. They, therefore, are demanding larger loans to boost their maize production and increase income. Moreover, the propensity of female farmers to secure larger loans compared to men can be attributed, in part, to the support and remittances they receive from their husbands. This support boosts their confidence and encourages them to seek substantial loan amounts. In some cases, these female farmers are de jure heads of household, with their husbands working in the cities. These females can take the loans in their names.

**Farmer type (Farmtype):** The results indicate that the A1 farmers were more likely to obtain a larger loan than the A2 farmers. Given that A1 farmers are very small (five hectares) compared to A2 farmers (larger than 36 hectares), these findings are contrary to previous findings [48–50,54,55], which found a positive relationship between farm size and farmers' participation in contract farming positively revealing that loan uptake by smallholder farmers surges with the proliferation in the size of the arable land they own.

According to the findings of this study, an increase in A1 farmers increases the odds ratios of them participating in command agriculture by 12%. A1 farmers' likelihood of acquiring larger loan sizes compared to A2 farmers is influenced by several factors. These include the fact that they have limited resources, face challenges in accessing credit from formal banks due to a lack of collateral, and lack of market linkages. Furthermore, farm income-diversification aspirations and capacity-building requirements also drive A1 farmers to seek larger loans to support their agricultural businesses and take advantage of the affordable command agricultural loans.

**Training in command agriculture:** The coefficient is −0.24, implying that farmers who did not receive training on command agriculture are likely to acquire larger loans compared to farmers who received adequate training. An increase in command education (information dissemination) is likely to decrease the loan sizes demanded by farmers by 24 percent. The findings were unexpected.

These are contrary to the findings by [56], who asserted that insufficient dissemination of information by lending companies is the main problem that limits participation in the contract schemes. This is mainly because training in command agriculture provides farmers with detailed and comprehensive information about the loan, significantly influencing farmers' participation by enhancing their awareness, understanding, and confidence. Training also motivates farmers to borrow significant amounts based on the demonstrated benefits and achieve their farming objectives [57–63]. In this study, the negative relationship between the command loan size and command loan training implies that the training that the farmers were getting was not relevant to financial management, risk mitigation strategies, and encouragement to obtain substantial loans. In addition, the negative relationship can be explained by the fact that trainers might have trained farmers more on the risks involved in participation than the benefits; therefore, the farmers feared borrowing the loan [64,65].

## 5. Conclusions

Using a sample of 350 smallholder farmers, this study implemented a double-hurdle model to determine factors affecting farmers' decisions to participate in command agriculture. It was found that gender, family size, farmer type, command agriculture education, and distance from the market influenced smallholder farmers' participation in command agriculture. The study also shows that participation in the command agriculture programme is an undeniable opportunity for smallholder farmers in the study area, especially female farmers, who are acquiring larger loans—revealing their passion for large-scale maize production. This also ensures farmers' food security since maize is a staple food in Zimbabwe. The unique contribution of this study is that it offers a more comprehensive analysis of both the decision-making process leading up to contract participation and the intensity of participation in contract farming. Therefore, policymakers should consider these factors to enhance and improve participation by smallholder farmers in command agriculture. Extension services also should intensify their training for female farmers, as well as encourage them to participate in command agriculture.

**Author Contributions:** Conceptualization, S.Z.; Validation, T.M.; Investigation, T.M.; Writing—review and editing, T.M., A.M. and S.Z.; Supervision, A.M.; Project administration, A.M.; Funding acquisition, S.Z. All authors have read and agreed to the published version of the manuscript.

**Funding:** We would like to thank the Govan Mbeki Research and Development Centre and the Zimbabwean Presidential Scholarship for funding the writing of this work.

**Institutional Review Board Statement:** Not applicable.

**Data Availability Statement:** The datasets used and analysed during the current study are available from the corresponding author upon reasonable request.

**Acknowledgments:** We would like to thank Saul Ngarava for assisting with revisions and parts of the analysis and the enumerators who helped in conducting farmer interviews.

**Conflicts of Interest:** The authors declare no conflict of interest.

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
