# Peer review of "Factors Influencing Smallholder Farmers’ Decisions to Participate in Loan-Based Farming in Mutare District, Zimbabwe—A Double-Hurdle Model Approach"

_agriculture, doi:10.3390/agriculture13122225_

Round 1
Reviewer 1 Report
Comments and Suggestions for Authors
Contract farming helps strengthen the resilience of smallholder farmers. The authors' focus on the factors that influence smallholder farmers' participation in contract farming in Zimbabwe, which is an interesting study. However, before considering publication, I think the authors need to do some revision work. As follows:
(1) When sampling, the author mentioned that 480 households were sampled, but 350 households were used in the research. I suggest the authors add why 130 households are missing.
(2) There is no Li in equation (1). I suggest that the author check whether the formula is written incorrectly.
(3) It is confusing that the authors study the factors that influence contract farming, but measure contract farming by whether farmers have access to loans. Loans are a poor representation of contract farming. I suggest that authors change the topic or change the measurement method.
(4) In the conclusion, the authors mention that "Command Agriculture loan can play a direct role in improving maize production and indirectly supports the agriculture sector's growth and development through the pro-vision of modern inputs." However, this conclusion has not been demonstrated in the empirical study, and I suggest that the authors make relevant conclusions based on the results of the study.
Author Response
Thank you very much for taking the time to review our manuscript. Please find the detailed responses below and the corresponding revisions in track changes in the submitted file.
Comment (1). When sampling, the author mentioned that 480 households were sampled, but 350 households were used in the research. I suggest the authors add why 130 households are missing.
Authors response: Although we have received 480 completed questionnaires, after the data cleaning process, a total of 350 questionnaires were suitable for analysis. The other 130 had many missing observations and we decided to drop those respondents.
Comment (2). There is no Li in equation (1). I suggest that the author check whether the formula is written incorrectly.
Authors' response: We have now corrected the formulas by ensuring that subscripts are clearly shown as subscripts e.g. 'Li' not as 'Li'
Comment (3). It is confusing that the authors study the factors that influence contract farming, but measure contract farming by whether farmers have access to loans. Loans are a poor representation of contract farming. I suggest that authors change the topic or change the measurement method.
Authors' response: Thank you for this suggestion. We have now changed the title to: " Factors influencing smallholder farmers' decision to participation in loan-based farming in Mutare District, Zimbabwe -A Double Hurdle model approach".
Comment (4). In the conclusion, the authors mention that "Command Agriculture loan can play a direct role in improving maize production and indirectly supports the agriculture sector's growth and development through the pro-vision of modern inputs." However, this conclusion has not been demonstrated in the empirical study, and I suggest that the authors make relevant conclusions based on the results of the study.
Authors response: We have removed that sentence and inserted new sentences where we conclude based on our study's objective and what we have found and also highlighted our study's unique contribution, please refer to the 'conclusion' section.
Reviewer 2 Report
Comments and Suggestions for Authors
The paper describes an interesting loan scheme applied in Zimbabwe and tries to analyse the variables influencing participation. As such this can really make up an interesting paper but to be published the paper misses a good theoretical and conceptual model behind the analysis. Therefore the authors have to write at different occasions that the influence of a variable is unexpected. So my recommendations are:
- To develop and describe the theoretical arguments why some variables should influence both the decision to participate and why the same (or other variables) should also influence the size of the participation. Further this framework should also explain why other variables are not taken up
- How is the dependent variable in the second hurdle model measured. I wonder whether using an absolute number is not introducing a bias.
- The table 1 and 2 should also reveal the variable averages (and eventually standard deviation) between participating and non-participating farmers per A1 and A2 group (with maybe already a statistical analysis of the differences)
- The model should be checked on whether the variables are really independent and on whether you can use the same variables in both hurdles, in particular when using for the second hurdle also binary variables (so check with a specialist in econometric models)
- The interpretation of the results should be much more carefully made given that they are often counter-intuitive and opposed to what is found by other authors. If the results are reliable, the interpretation and explanation should be much stronger.
For the rest see my comments in the annotated paper to see where the problems are situated.

see annotated file
Author Response
Thank you very much for taking the time to review our manuscript. Please find the detailed responses below and the corresponding revisions in track changes in the re-submitted files.
Comment (1). To develop and describe the theoretical arguments why some variables should influence both the decision to participate and why the same (or other variables) should also influence the size of the participation. Further this framework should also explain why other variables are not taken up.
Authors' response: We have inserted a new section (2), where we use Adam Smith's theory to explain how the demand for Command Agriculture loan plays out among our respondents. We also highlight why the variables we have used influence the decision to take up loans among our respondents.
Comment (2). How is the dependent variable in the second hurdle model measured. I wonder whether using an absolute number is not introducing a bias.
Authors' response:
The dependent variable in the second hurdle is loan size and is specified in the revised version of the paper. The second hurdle as shown in Table 4 does not refer to hectares but rather the size of the loan that is taken out. Please refer to Ogouve et al. (2022) as well as Anang and Dagunga (2023) who modeled the second hurdle as the one used in the study. Equally also, the study used the same model.
Ogouvide et al (2022), Determinants of farmer's participation in formal microcredit markets in Benin: A double hurdle model, Journal of Academic Finance, 13(2), 77-9. https://doi.org/10.59051/joaf.v13i2.610
Anang B. T. and Dagunga G. (2023), Farm household access to agricultural credit in Sagnarigu Municipal of Northern Ghana: Application of Gragg's Double Hurdle Model, 1-12, https://doi.org/10.1177/00219096231154234
Comment (3). The table 1 and 2 should also reveal the variable averages (and eventually standard deviation) between participating and non-participating farmers per A1 and A2 group (with maybe already a statistical analysis of the differences).
Authors' response: We have revised the Table 3 and included statistical analysis of the variables used and differences between A1 and A2 farmers.
Comment (4). The model should be checked on whether the variables are really independent and on whether you can use the same variables in both hurdles, in particular when using for the second hurdle also binary variables (so check with a specialist in econometric models).
Authors' response: Comment noted and appreciated. The objective of the study is not to compare the two groups of farmers, but rather what affects loan acquisition and the loan size. It's actually better to include farm category as an independent variable, which might however introduce multicollinearity.
Comment (5). The interpretation of the results should be much more carefully made given that they are often counter-intuitive and opposed to what is found by other authors. If the results are reliable, the interpretation and explanation should be much stronger.
Authors' response: Thank you for this valuable observation. We have now backed our analysis with priori theory, see the inserted part on the methodology section where we also included Table 1, showing expectations on how the independent variables could influence the dependent variable. Further in the analysis itself, we have added few sentences to strengthen the discussion citing literature.
Round 2
Reviewer 1 Report
Comments and Suggestions for Authors
I am satisfied with the revised manuscript.